# Interspatial Distribution of Tumor and Immune Cells in Correlation with PD-L1 in Molecular Subtypes of Gastric Cancers

**DOI:** 10.3390/cancers14071736

**Published:** 2022-03-29

**Authors:** Bastian Dislich, Kirsten D. Mertz, Beat Gloor, Rupert Langer

**Affiliations:** 1Institute of Pathology, University of Bern, 3008 Bern, Switzerland; 2Institute of Pathology, Cantonal Hospital Baselland, 4410 Liestal, Switzerland; kirsten.mertz@ksbl.ch; 3Department of Visceral Surgery and Medicine, Inselspital Bern, University of Bern, 3010 Bern, Switzerland; beat.gloor@insel.ch; 4Institute of Clinical Pathology and Molecular Pathology, Kepler University Hospital, Johannes Kepler University, 4021 Linz, Austria; rupert.langer@kepleruniklinikum.at

**Keywords:** gastric cancer, PD-L1, TIL, EBV, MMR, MSI, immunotherapy

## Abstract

**Simple Summary:**

The majority of gastric cancers are negative for the Epstein–Barr virus (EBV) and mismatch repair-proficient, or microsatellite stable. Molecular subtyping currently does not influence clinical decision making, and accurate response prediction towards immunotherapy remains a major challenge. We therefore analyzed PD-L1 expression, tumor-infiltrating lymphocytes, and their spatial relationship to tumor cells in EBV-negative mismatch repair-proficient gastric cancers compared to EBV-positive and mismatch repair-deficient tumors to identify an immunogenic phenotype that is susceptible to immunotherapy in this large group. We demonstrated a close relationship between the total number of tumor-infiltrating lymphocytes, their proximity to tumor cells, and the expression of PD-L1 across all subtypes, including the EBV-negative and mismatch repair-proficient cancers. However, we also identified a subgroup of PD-L1-negative, EBV-negative, and mismatch repair-proficient cancers with high numbers of tumor-associated CD8+ lymphocytes. This is indicative of an immunoreactive phenotype in a subgroup of gastric cancers along or independent of PD-L1 status and molecular type.

**Abstract:**

(1) Background: EBV-positive and mismatch repair-deficient (MMRd) gastric cancers (GCs) show higher levels of tumor-infiltrating lymphocytes (TILs) and PD-L1 expression and thus a more profound response to immunotherapy. However, the majority of GCs are EBV-negative (EBV−) and MMR proficient (MMRp). We analyzed PD-L1 expression and TILs in EBV-MMRpGCs in comparison to EBV-positive (EBV+) and MMRdGCs to identify an immunogenic phenotype susceptible to immunotherapy. (2) Methods: A next-generation tissue microarray of 409 primary resected GCs was analyzed by Epstein-Barr encoding region (EBER) in situ hybridization for MSH1, PMS2, MSH2, MSH6, PD-L1, and CD8 immunohistochemistry. PD-L1 positivity was defined as a combined positive score (CPS) of ≥1. CD8+ TILs and their proximity to cancer cells were digitally analyzed on the HALO™ image analysis platform. (3) Results: Eleven cases were EBV+, 49 cases MMRd, and 349 cases EBV-MMRpGCs. The highest rate of PD-L1 positivity was seen in EBV+GCs, followed by MMRdGCs and EBV-MMRpGCs (81.8%, 73.5%, and 27.8%, respectively). EBV+ and MMRdGCs also demonstrated increased numbers and proximity of CD8+ TILs to tumor cells compared to EBV-MMRpGCs (*p* < 0.001 each). PD-L1 status positively correlated with the total numbers of CD8+ TILs and their proximity to tumor cells in all subtypes, including EBV-MMRpGCs (*p* < 0.001 each). A total of 28.4% of EBV-MMRpGCs showed high CD8+ TILs independent of PD-L1. (4) Conclusions: PD-L1 and CD8 immunohistochemistry, supplemented by digital image analysis, may identify EBV-MMRpGCs with high immunoreactivity indices, indicating susceptibility to immunotherapy.

## 1. Introduction

Gastric cancer (GC) is the fifth most common cancer and the third most common cause of cancer-related death worldwide. The overall median survival, with an average of 50 months, remains low, despite using modern therapy regimens [1]. Current research focuses on translating the recent understanding of GC molecular biology, in particular the now established molecular subtypes, into more tailored therapeutic regimens. In addition, immunotherapy, which has emerged as a powerful therapeutic option in many solid cancers, is an emerging therapeutic option for a subset of GCs [2,3]. The strong relationship between the molecular subtypes, the response to immune checkpoint inhibition, and the related presence or absence of predictive biomarkers, will likely result in a multidimensional assessment of GCs for tailored therapeutic regimens in the near future [4]. The molecular subtypes resulted from an integrative genomic and transcriptomic analysis by the Cancer Genome Association (TCGA) consortium and Asian Cancer Research Group (ACRG) in 2014 and 2015, respectively [5,6]. The four major subtypes according to TCGA nomenclature are Epstein–Barr virus-positive (EBV+) GCs, microsatellite instable GCs, chromosomal instable GCs, and genomic stable GCs. In routine diagnostics, EBV positivity is determined by the direct detection of EBV RNA by EBER in situ hybridization, and MSI is determined by either PCR or the detection of mismatch repair deficiency (MMRd) by immunohistochemistry. EBV+ and MMRd GCs subgroups are associated with a better prognosis and an increased response to anti-PD1 (programmed cell death protein 1)/anti-PD-L1 (programmed death-ligand 1) immune checkpoint inhibition. This is due to the amplification of the PD-L1 gene, a pro-inflammatory tumor microenvironment in the former subtype, and a high tumor mutational burden in the latter subtype [2,3]. These subtypes thus directly influence clinical decision making. In contrast, the far more common EBV-negative (EBV−) and mismatch repair-proficient (MMRp) chromosomal instable and genomic stable GC subtypes currently do not influence clinical management, and accurate response prediction remains a major challenge [1,7,8,9].

As the PD-L1 protein drives the tumor immune escape and serves as a target for immune checkpoint inhibitors, its expression in cancer tissue is an established biomarker for response prediction [8]. Regardless of the molecular subtype, patients with previously treated advanced gastric cancer with a PD-L1 combined positive score (CPS) ≥ 1 show a significantly higher overall response rate to immune checkpoint inhibitors than PD-L1-negative patients. However, the overall objective response rate to immune checkpoint inhibition in GCs is low (12%), and PD-L1 expression in GCs is an imperfect and controversial biomarker [10,11]. As immune-based therapies are highly intricate and dynamic, multidimensional biomarkers could improve response prediction. Thus, in addition to PD-L1 expression, other predictive biomarkers for immunotherapy are under investigation, including tumor mutational burden, immune-related gene expression analysis, circulating tumor DNA, and the tumor microenvironment, including tumor-infiltrating lymphocytes (TILs) [8,10,12,13]. One of these multidimensional approaches uses a simplified model of the tumor microenvironment that relies on PD-L1 expression and the number of TILs, and has been applied to various solid malignancies, including GCs and esophageal adenocarcinoma [14,15,16]. Furthermore, not only the quantity but also the quality of the interaction between tumor cells and immune infiltrates may have a biologic impact. Analysis of the spatial distribution of tumor and immune cells and the vicinity of these two compartments could offer novel insights into the complex field of tumor–host interaction.

In this study, we analyzed PD-L1 expression, immune infiltrates, and their spatial relationship to tumor cells in EBV-MMRpGCs in comparison to EBV+ and MMRdGCs to evaluate whether alternative characteristics beyond EBV and MSI status may indicate an immunogenic phenotype susceptible to immunotherapy. Immunohistochemical stainings for the characterization of both immune infiltrates and cancer tissue were applied on a next-generation tissue microarray described recently in a previous study [17]. Quantification of the immune infiltrates and analysis of the spatial distribution of tumor and immune cells were performed using digital image analysis. The results were correlated with the PD-L1 status and the molecular subtypes.

## 2. Materials and Methods

Cohort and tissue microarray: The study used buffered formalin-fixed paraffin-embedded tissue from 409 human gastric adenocarcinomas of patients that underwent primary resection for gastric cancer at the Department of Surgery, Inselspital Bern, University of Bern, between 1993 and 2013. Tumors of all pT categories and pN categories, as well as the molecular subtypes of EBV+, MMRd, and EBV-MMRpGCs, were included. A next-generation tissue microarray (ngTMA) of three tissue cores (core size 0.6 mm) each of the tumor center and the tumor front of the resection specimen was constructed as previously described [18]. The TMA and the cohort, as well as the molecular characterization of the tumors, are described in more detail in a previous manuscript [17].

Immunohistochemistry and PD-L1 scoring: Immunohistochemistry for PD-L1 (clone SP263, Ventana, Roche Diagnostics, Oro Valley, AZ, USA) was carried out on an automated immunostainer (Ventana BenchMark GX) according to the manufacturer’s recommendation. Double-staining immunohistochemistry for pan-cytokeratin (clone AE1/AE3, Dako, Agilent, Glostrup, Hovedstaden, Denmark) and CD8 (clone 4B11, Novocastra, Leica Biosystems, Nussloch, Germany) was performed using mixed DAB Refine and Mixed Red reagents (Leica Biosystems) on an automated immunostainer (BOND-III, Leica Biosystems) according to the manufacturer’s recommendation. All samples were counterstained with hematoxylin and mounted in Aquatex (Merck, Darmstadt, Germany). PD-L1 scoring was performed by one board-certified pathologist (BD) across all cores of one individual tumor using a conventional microscope and the combined positive score (CPS) scoring system. The CPS score was calculated using the number of tumor cells with linear membrane staining, plus the number of lymphocytes and macrophages with a cytoplasmic or membrane staining divided by the total number of viable tumor cells, multiplied by 100. Cases with an overall CPS of ≥1 were considered positive, as this corresponds to the cutoff value for immune checkpoint inhibitor monotherapy to demonstrate a survival benefit in a recent meta-analysis [19]. Additionally, the cutoff of a CPS ≥ 5 was used in the descriptive analysis due to the results of the CheckMate 649 trial [11].

Digital image analysis: The mounted glass slides were scanned with a Philips IntelliSite Ultra-Fast Scanner (Philips, Koninklijke Philips N.V., Amsterdam, The Netherlands) and converted to digital whole-slide images. Detection of positive stained cells and spatial analysis were performed on the HALO^TM^ digital image analysis platform (HALO V2.0, Indica Labs, Albuquerque, NM, USA). CD8+ lymphocytes and tumor cells were registered by the multiplex IHC module using color deconvolution and nuclear segmentation of the differentially stained cells. The results of the cell registration were manually reviewed for all TMA cores; false positive annotations (e.g., pan-cytokeratin-positive benign epithelium or stroma, intravascular tumors, or lymphocytes) or cores with technical artefacts were excluded from the analysis. The proximity of the CD8+ lymphocytes to the tumor epithelium was analyzed with the spatial analysis module and recorded for various distances (1, 5, 20, and 100 μm). The mean number of lymphocytes per TMA core of the tumor center was reported (maximum three cores/case). The mean number of all lymphocytes in the TMA core is referred to as CD8^all^, and the mean number of intratumoral lymphocytes (within or ≤1 μm from tumor tissue) is referred to as CD8^int^.

Statistical analysis: Statistical analysis was carried out using the IBM SPSS 28.0 Statistics software (IBM, Chicago, IL, USA) and Microsoft Excel (Microsoft, Redmond, WA, USA). Correlations between categorical variables were conducted using χ-square and Fisher’s exact tests. *p*-values were two-sided and regarded as significant if *p* < 0.05.

## 3. Results

All 409 tumors were grouped according to their EBV and MMR status, which was mutually exclusive for all cases, into three subtypes: EBV+, MMRd, and EBV-MMRp. A total of 11 out of 409 (2.7%) tumors were EBV+, as determined by EBER in situ hybridization, 49 of 409 (12%) were MMRd, with concomitant loss of MLH1 and PMS2 expression by immunohistochemistry, and 349 of 409 (85.3%) were EBV-MMRp. The number of CD8+ intratumoral and peritumoral lymphocytes and their distance to the tumor were recorded for each TMA core, as described above. Representative images of a tumor stained by double immunohistochemistry for CD8 and pan-cytokeratin, as well as the digital overlay after image analysis, is shown in Figure 1A,B. The distribution of all intra- and peritumoral CD8+ lymphocytes (CD8^all^) or only intratumoral CD8+ lymphocytes (intraepithelial or immediately adjacent, i.e., ≤1 μm from tumor epithelium; CD8^int^) across the three subtypes is shown in Figure 1C,D.

Using the median of CD8^all^ or CD8^int^ as a cutoff, EBV+GCs showed the strongest correlation with a high number of TILs. A total of 90.9% of cases were CD8^all^ high (*p* = 0.006), and 100% of cases were CD8^int^ high (*p* = 0.002). MMRdGCs were also associated with a high number of TILs for CD8^all^ (69.4%, *p* = 0.005) and CD8^int^ (83.7%, *p* < 0.001), but to a lesser degree than EBV+GCs. As it is evident from the percentage of EBV+ and MMRd cases within the CD8^all^ high and CD8^int^ high categories, the CD8^int^ high status showed a stronger correlation than the CD8^all^ high status for the EBV+ and MMRd subtypes of GCs. This was also evident when analyzing the CD8+ TILs within various distances from the tumor epithelium (100 μm, 20 μm, and 5 μm); each increase in proximity of the CD8+ lymphocytes to the tumor epithelial cells showed a stronger association with the EBV− and MMRd GC subtypes. For EBV-MMRpGCs, approximately half of the cases belonged to the CD8^all^ high (46.4%) and CD8^int^ high (47.3%) categories (Figure 2).

In line with the higher number of intra- and peritumoral CD8+ lymphocytes, both EBV+ and MMRdGCs were correlated with a positive PD-L1 status (a CPS ≥ 1; 81.8% and 73.5% respectively, *p* < 0.001 each). For MMRpGCs, only 27.8% of cases were PD-L1-positive (Figure 3A). In addition, PD-L1-positive tumors demonstrated increased numbers of CD8+ TILs with increased proximity to tumor cells compared to PD-L1-negative tumors across all subtypes (*p* < 0.001 for CD8^all^, CD8^int^, and within all other distances; Figure 3C). As the results of the Checkmate 649 trial suggested a CPS of ≥5 as the most meaningful cutoff for PD-L1 positivity in terms of response prediction to combined immune checkpoint inhibition and chemotherapy, we additionally calculated how this cutoff influences the number of PD-L1-positive tumors among the different subtypes [11]. EBV+ and MMRdGCs were still correlated with a positive PD-L1 status (*p* < 0.001 each), but the number of PD-L1-positive cases decreased to 12.3% for MMRpGCs, 40.8% for MMRdGCs, and 63.6% for EBV+GCs (Figure 3B).

The CD8 status and PD-L1 expression were correlated with clinicopathological parameters. High numbers of tumor-infiltrating lymphocytes were observed more often in cancers with lower pT categories (≤T3; *p* = 0.033 for CD8^all^ and *p* = 0.001 for CD8^int^) and without lymph node metastases (*p* = 0.045 for CD8^int^). High numbers of tumor-infiltrating lymphocytes and PD-L1 expression were more often observed in cancers with intestinal-type morphology based on the Laurén classification (*p* = 0.022 for CD8^int^; *p* = < 0.001 for a CPS ≥ 1). The detailed correlations of the PD-L1 and CD8 status are shown in Table 1.

Using the PD-L1 status and the amount of tumor-infiltrating lymphocytes, the tumor microenvironment of solid cancers can be classified into four different immunological subgroups that reflect their immune status [16,20]. In analogy to this proposed scheme, we assigned all cases of the EBV+, MMRd, and EBV-MMRpGCs to one of the four subgroups based on their PD-L1 status and CD8^int^ category: type I (adaptive immune resistance; PD-L1+/CD8^int^ high), type II (immunological ignorance; PD-L1-/CD8^int^ low), type III (intrinsic induction; PD-L1+/CD8^int^ low), and type IV (tolerance; PD-L1-/CD8^int^ high) (Figure 4). EBV+ and MMRdGCs predominantly showed a type I tumor microenvironment, whereas EBV-MMRpGCs most often showed a type II tumor microenvironment.

## 4. Discussion

In this tissue-based exploratory study, we analyzed a large cohort of primary resected gastric cancers, including EBV-associated gastric carcinomas (EBV+GCs), mismatch repair-deficient gastric carcinomas (MMRdGCs), and EBV negative mismatch repair-proficient gastric carcinomas (EBV-MMRpGCs). We used a combination of immunohistochemical stainings (PD-L1, pan-cytokeratin, and CD8) and digital image analysis to characterize the immune infiltrates with a focus on tumor-infiltrating lymphocytes (TILs) and the proximity of T cells to the next located tumor cell as sign of an active relation between the tumor and the host. We could demonstrate that (a) EBV+GCs and MMRdGCs show a higher degree of TILs, in particularly close peritumoral/intraepithelial CD8+ lymphocytes; these tumors also more frequently showed a positive PD-L1 status, as determined by a CPS ≥ 1; (b) PD-L1 status correlated in general with higher immune infiltrates and a closer spatial relationship of the immune cells and the cancer cells; and (c) a significant subset of EBV-MMRpGCs showed PD-L1 positivity and/or high peritumoral/intraepithelial CD8+ lymphocytes.

The combined expression of PD-L1 on tumor tissue and associated mononuclear inflammatory cells is used as a predictive biomarker for immunotherapy and reported via the CPS scoring system for various solid malignancies. Based on the phase 3 CheckMate-649 trial, the Food and Drug Administration recently approved the anti-PD-1 inhibitor nivolumab in combination with chemotherapy for the initial treatment of patients with advanced GCs. The addition of immunotherapy over chemotherapy alone resulted in a significant improvement in overall survival in patients with a PD-L1 CPS ≥ 1 [11]. A CPS of ≥1 was also suggested in a recent meta-analysis as the cutoff value where immune checkpoint inhibitor monotherapy shows a survival benefit [19]. A total of 35% of all GC cases of our cohort were PD-L1-positive (i.e., a CPS ≥ 1), and PD-L1 positivity was associated with EBV+ and MMRd subtypes. In line with our results, higher PD-L1 levels have been detected in EBV+GCs in particular. The molecular rationale between these associations is based on at least two observations. First, a subset of EBV+GCs show an amplification of the chromosome 9p24.1, which includes the genes for PD-L1 and PD-L2 [5]. Second, and independent of 9p24.1 amplification, EBV+GCs show a transcriptomic landscape related to enhanced T cell cytotoxic function, as well as enhanced interferon gamma and pro-inflammatory cytokine signaling [21,22]. Current data suggest that EBV infection promotes an inflamed tumor microenvironment that is associated with an increased response to targeted immunotherapies [10]. Although less strong than for EBV+GCs, we also observed a correlation of PD-L1 expression with MMRdGCs in our cohort, which also reflects the results of previous studies [23,24]. The loss of MMR activity leads to the accumulation of single nucleotide variants and frameshift mutations, and thus numerous subsequently neoepitopes and a high tumor mutational burden. This molecular phenotype is accompanied by an increased number of TILs, tumoral PD-L1 expression and response to immune checkpoint inhibitors [25]. However, in a recent prospective study, approximately half of MMRdGCs were shown to be resistant to immunotherapy, and there is no clear association of PD-L1 expression and outcome for this GC subtype [26]. Thus, besides examining the PD-L1 status in MMRdGCs, classifying those cases into CD8^int^ high or low might be beneficial for the identification of putative responders to immunotherapy. PD-L1 positivity in our cohort, however, was not restricted to EBV+ and MMRdGCs, as close to one third of the EBV-MMRpGC cases were PD-L1-positive. In general, PD-L1 positivity was associated with higher TIL counts, which is in line with previous studies [20,27,28]. When choosing a CPS of ≥5 as a cutoff for maximum clinical efficacy for immune checkpoint inhibition, as discussed in the study of the CheckMate 649 trial, the association of PD-L1 positivity with EBV+ and MMRdGCs is still observed, but the percentage of PD-L1-positive cases in this subgroup drops from 75% to 60%, arguing that for our cohort, a cutoff at a CPS of ≥1 for PD-L1 positivity better reflects the inherent immunogenic signature of these subtypes.

Higher numbers of TILs and their proximity to tumor cells were associated with EBV+ and MMRdGCs and more favorable clinicopathological parameters (lower pT and pN categories). The correlation of the proximity of the peritumoral CD8+ lymphocytes with the molecular subgroups and the PD-L1 status indicates that the lymphocytes close to the tumor epithelium are the most functionally relevant. This is plausible, as the cytotoxic function and signaling of CD8+ lymphocytes requires their proximity to their target cells and is supported by a statistically stronger association of the intra- vs. peri- and intratumoral CD8+ lymphocytes with lower pT and pN categories in our study. The relevance of this spatial relationship has been shown in different tumors, including GCs, and is supported by our data [29,30]. As PD-L1 itself is an imperfect biomarker for response prediction in GCs, the analysis of the tumor microenvironment, including TILs, has been investigated in a few studies. One classification scheme uses the T-cell density and the PD-L1 status to classify solid cancers (including GCs) into four tumor microenvironment types [16,20]. In this paper, we show that the vast majority of EBV+ and most MMRdGCs belong to type I (adaptive immune resistance), whereas most of the EBV-MMRpGCs belong to type II (immunological ignorance), which is in agreement with previous studies [20,31,32]. As most of the EBV+ GCs that were analyzed in this study belong to tumor microenvironment type II, one could speculate whether the determination of the EBV status by the cost-intensive EBER in situ hybridization could be omitted in routine diagnostics, as those cases would be classified as type I by their CD8^int^ high and positive PD-L1 expression, and thus would be most likely assigned to the category of tumors that benefit most from single-agent anti-PD-1/L1 therapy [16]. We also observed EBV-MMRpGCs with high numbers of intra- and/or peritumoral CD8+ lymphocytes, thus reflecting a pronounced immune response of the host towards tumor tissue. Our data indicate that in this largest subtype of GCs, at least a subset of cases (corresponding to tumor microenvironment types I and IV) may be eligible for immunotherapy [16,33].

Our work suffers from several limitations. First, despite including more than 400 cases of GC, the total number of EBV+GCs is low in our cohort, due to the rarity of this molecular subtype. Second, the percentage of PD-L1-positive cases (i.e., a CPS ≥ 1; 35%) in our cohort is lower than in comparable studies, where percentages between 46% and 83% are reported using similar or comparable PD-L1 detection assays [11,34,35]. Our PD-L1 assay was performed in an ISO-certified laboratory and regularly subjected to independent interlaboratory proficiency testing, which makes assay associated technical issues less likely. We can speculate on several reasons, including, but not limited to, interobserver variability, the TMA-based study design, and the use of archival tissue from a cohort dating back to 1993. Third, our study lacks a long-term follow-up, with missing overall survival data. However, the available clinicopathological parameters show a beneficial course of disease, e.g., for MMRdGCs with lower pT categories, lower histological grades, and less distant and locoregional metastases, indicating the well-known less aggressive behavior [17]. Moreover, the cases collection is from a period before the results of the MAGIC trial were published [36]. Since multimodal treatment of advanced GCs is nowadays considered standard therapy, outcome data from this cohort may not be comparable to current clinical practice. On the other hand, the investigation of primary resected, therapy-naïve GCs allows for the study of PD-L1 expression, TILs, and the clinical course without the potential confounders of perioperative chemotherapy.

## 5. Conclusions

Our data support a molecular and immune profile of EBV+ and MSI/MMRd gastric cancers with constantly higher immune cell infiltrates in these tumors. Here, we show that discrimination of these molecular subtypes can be improved using CD8+ TIL counts of the very nearly located, i.e., intraepithelial, T cells. We also confirmed the finding that these tumors more frequently show a positive PD-L1 status. It should be noted, though, that these two intrinsically “immune-hot” tumors only represent a small percentage of GCs, as the large majority of GCs are chromosomal instable or genomic stable tumors showing a molecular EBV-MMRp genotype. In this paper, we show that this group of GCs features a substantial number of cases with high numbers of intra- and/or peritumoral CD8+ lymphocytes, along or independent of the PD-L1 status, thus reflecting a pronounced immune response of the host towards tumor tissue. Our data indicate that in this largest subtype of GCs, at least a subset of it will be responsive to immunotherapy. Given the observation that some PD-L1-negative tumors also show high CD8+ TIL levels, we suggest that PD-L1 status alone may not completely reflect the immunogenicity of GCs or their susceptibility to immunotherapy.

## Figures and Tables

**Figure 1 cancers-14-01736-f001:**
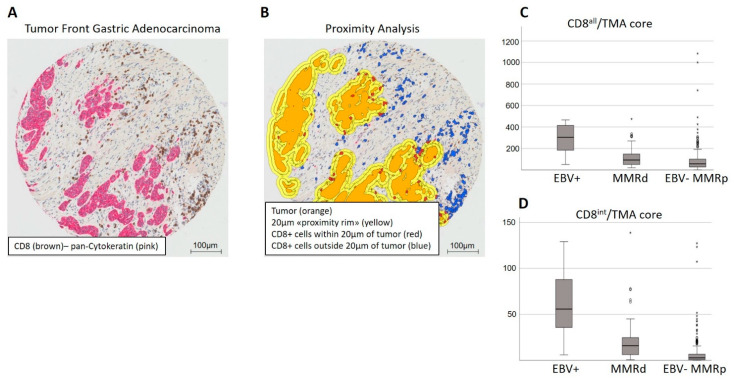
Digital image analysis of intra- and peritumoral CD8+ lymphocytes. (**A**) Double immunohistochemistry of a TMA core of gastric cancer. (**B**) Digital overlay of the same core after image analysis, demonstrating the accuracy of cell detection and tumor boundary. (**C**) Box plots demonstrating the distribution of all intraepithelial and peritumoral (CD8^all^) or (**D**) only intratumoral (CD8^int^) CD8+ lymphocytes per TMA core amongst the different GC subtypes. The circles and stars in (**C**,**D**) represent outliers. Circles represent outliers above the upper limit of quartile 3 by a factor of 1.5 times the interquartile range. Stars represent outliers above the upper limit of quartile 3 by a factor of 3 times the interquartile range.

**Figure 2 cancers-14-01736-f002:**
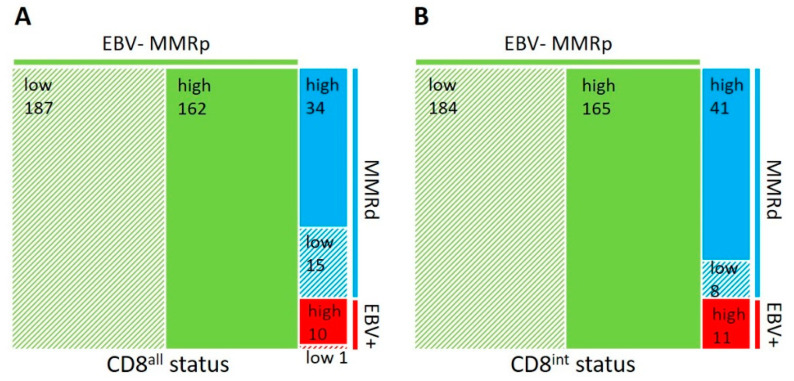
Correlation of tumor-infiltrating CD8+ lymphocytes with EBV and MMR status. Area proportional tree maps are shown based on the number of all (intra- and peritumoral) CD8+ lymphocytes (CD8^all^) (**A**) or intratumoral CD8+ lymphocytes (CD8^int^) (**B**) per TMA core. Note the increased association of the CD8^int^ high status with the EBV+ and MMRd subtypes in comparison to the CD8^all^ status.

**Figure 3 cancers-14-01736-f003:**
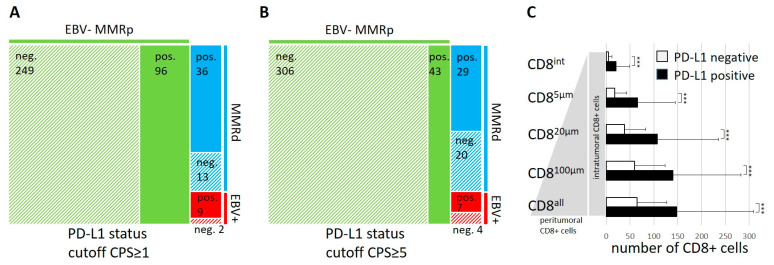
Correlation of PD-L1 expression with EBV and MMR status and tumor-infiltrating lymphocytes. An area proportional tree map is shown based on the PD-L1 status using a CPS ≥ 1 (**A**) or CPS ≥ 5 (**B**) as a cutoff. A bar graph demonstrates the association of a positive PD-L1 status (a CPS ≥ 1) with increased numbers of intratumoral and peritumoral CD8+ lymphocytes (**C**). *** refers to *p* ≤ 0.001.

**Figure 4 cancers-14-01736-f004:**
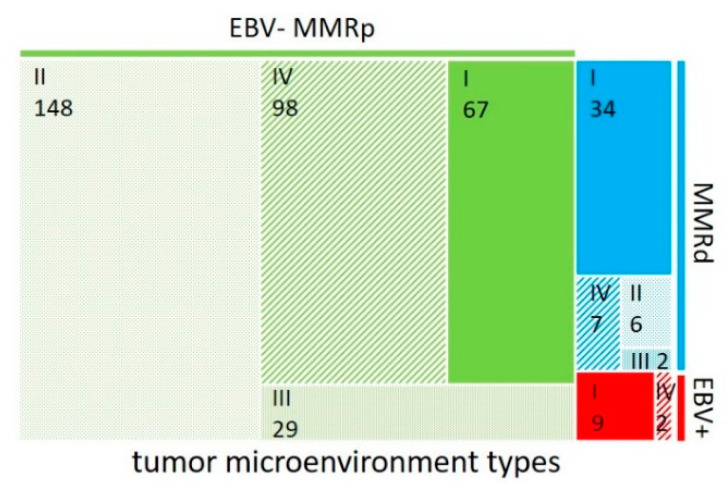
Distribution of tumor microenvironment types based on PD-L1 status and the number of intratumoral CD8+ lymphocytes across EBV+, MMRd, and EBV-MMRpGCs. An area proportional tree map illustrates the frequency of the different tumor microenvironment types depending on the EBV and MMR status of the tumors. Tumor microenvironment types are based on the PD-L1 and CD8^int^ status: type I (adaptive immune resistance; PD-L1+/CD8^int^ high), type II (immunological ignorance; PD-L1-/CD8^int^ low), type III (intrinsic induction; PD-L1+/CD8^int^ low), and type IV (tolerance; PD-L1-/CD8^int^ high).

**Table 1 cancers-14-01736-t001:** Correlations of PD-L1 and CD8 status with clinicopathological parameters.

Factors	n	CD8^all^ High	CD8^all^ Low	*p*-Value	CD8^int^ High	CD8^int^ Low	*p*-Value	n	CPS ≥ 1	CPS < 1	*p*-Value
pT				0.033			0.001				0.015
T1	45 (11%)	20 (44.4%)	25 (55.5%)		29 (64.4%)	16 (35.6%)		48 (11.7%)	16 (33.3%)	32 (66.7%)	
T2	54 (13.3%)	36 (66.7%)	18 (33.3%)		37 (68.5%)	17 (31.5%)		55 (13.5%)	28 (50.9%)	27 (49.1%)	
T3	146 (35.9%)	80 (54.8%)	66 (45.2%)		87 (59.6%)	59 (40.4%)		144 (35.3%)	54 (37.5%)	90 (62.5%)	
T4	162 (39.8%)	71 (43.8%)	91 (56.2%)		65 (40.1%)	97 (59.9%)		161 (39.5%)	44 (27.3%)	117 (72.7%)	
pN				0.37			0.045				0.112
N0	112 (27.5%)	61 (54.5%)	51 (45.5%)		69 (61.6%)	43 (38.4%)		112 (27.8%)	46 (41.1%)	66 (58.9%)	
N1-3	295 (72.5%)	146 (49.5%)	149 (50.5%)		149 (50.5%)	146 (49.5%)		291 (72.2%)	95 (32.6%)	196 (67.4%)	
Grading				0.278			0.107				0.041
G1	21 (5.2%)	8 (38.1%)	13 (61.9%)		14 (66.6%)	7 (33.3%)		23 (5.6%)	8 (34.8%)	15 (65.2%)	
G2	96 (23.6%)	45 (46.9%)	51 (53.1%)		58 (60.4%)	38 (39.6%)		97 (23.8%)	44 (45.4%)	53 (54.6%)	
G3	290 (71.2%)	154 (53.1%)	136 (46.9%)		146 (50.3%)	144 (49.7%)		288 (70.6%)	90 (31.3%)	198 (68.7%)	
Laurén				0.933			0.022				<0.001
Intestinal	207 (50.9%)	102 (49.3%)	105 (50.7%)		125 (60.4%)	82 (39.6%)		211 (51.7%)	101 (47.9%)	110 (52.1%)	
Diffuse	131 (32.2%)	69 (52.7%)	62 (47.3%)		58 (44.3%)	73 (55.7%)		127 (31.1%)	22 (17.3%)	105 (82.7%)	
Mixed	65 (16%)	34 (52.3%)	31 (47.7%)		32 (49.2%)	33 (50.8%)		66 (16.2%)	16 (24.2%)	50 (75.8%)	
Indeterm.	4 (0.9%)	2 (50%)	2 (50%)		3 (75%)	1 (25%)		4 (1%)	3 (75%)	1 (25%)	

## Data Availability

There is no publicly archived dataset for this study.

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
