# Peer review of "Interspatial Distribution of Tumor and Immune Cells in Correlation with PD-L1 in Molecular Subtypes of Gastric Cancers"

_cancers, 2022, doi:10.3390/cancers14071736_

Round 1

Reviewer 1 Report

The authors have addressed all questions from the reviewers satisfactorily. They have made adjustments in the manuscript accordingly. 

Reviewer 2 Report

Dear Authors,

I read the authors' response to my comments about this manuscript.   Although the first request of molecular analysis about tumor infiltrating lymphocytes and PDL-1 expression was not possible, I understand about it in reality.   However,  the authors gave us an effort to meet the 2nd comments by adding the cancer staging information, so I think this manuscript  is better than the previous one.   Thank you

This manuscript is a resubmission of an earlier submission. The following is a list of the peer review reports and author responses from that submission.

Round 1

Reviewer 1 Report

The manuscript by Dislich et al. describes the presence of CD8+ and PD-L1 positivity in molecular subtypes of gastric cancers. The authors pose that a combination of high number of TIL and CPS ≥ 1 may identify a group of EBV-MMRp gastric cancer patients that could potentially benefit from immune checkpoint inhibition.

Comments:

  • Gastric carcinomas are known for their intra-tumoral heterogeneity. Next generation TMA’s were used for CD8 and PD-L1 analysis. Could the authors discuss how this may have affected the results?
  • The intensity of immunohistochemical stainings may vary between different tumors or TMA cores. Were the settings for the digital image analysis (color deconvolution and segmentation) the same for all tumors/cores?
  • The authors hypothesized that characteristics of EBV+ and MMRd may serve as example to identify which of the EBV-MMRp tumors may also respond to immunotherapy. However, for definition of CD8 high vs low they used the median CD8+ cell densities in all tumors. Most likely still most of the CD8+ high EBV-MMRp tumors have much lower CD8+ densities than EBV+ or MMRd cases. It is unclear why the authors did not use the EBV+ and MMRd cases to define a cut-off for CD8+ high and low.
  • Overall, 34.8% of cases were PD-L1 positive (CPS≥1), irrespective of molecular subtype. Could the authors discuss why this percentage is much lower than the 83% CPS≥1 in the Checkmate-649 trial?
  • In advanced gastric cancer CPS≥5 has been used has been used as cut-off for efficacy of immunotherapy (Checkmate 649). Recently, Zhao et al showed a lack of benefit in the group of patients with CPS1-4. Why did the authors used a cut-off of CPS≥1? What would be the impact of this cut-off on the results? Could they add the data with CPS≥5 as cut-off?

Reviewer 2 Report

As the development of immune therapy for gastric cancers, PD-L1 expression can be an important factor for the oncologist to have a choice of the treatment method for advanced gastric cancer.  

This is the interesting research of immune related gene expression analysis for tumor microenvironment which investigate the relationship between the number of tumor infiltrating lymphocytes (TIL) and PD-L1 expression.

However, the overall PD-L1 expression and response rate to immune therapy in gastric cancer is relatively low and controversial, so I would like to point out some additional points to be taken in order to be a better study.

In this tissue based exploratory study, they used a combination of immunohistochemical staining and digital image analysis, but it would be better to provide detailed information on chromosome maps and nucleotide sequences in gastric cancer tissues for a more accurate analysis of the correlation between PD-L1 expression and tumor lymphocytes by the next generation sequencing method.

Also, the authors analyzed a large cohort of primary resected gastric cancers, but they did not provide an available clinicopathological parameters and long-term follow up survival data including tumor and lymph nodal stage and histological grade which reflect the aggressiveness of gastric cancer and treatment response.
